# Alteration of 3D Matrix Stiffness Regulates Viscoelasticity of Human Mesenchymal Stem Cells

**DOI:** 10.3390/ijms22052441

**Published:** 2021-02-28

**Authors:** Ting-Wei Kao, Arthur Chiou, Keng-Hui Lin, Yi-Shiuan Liu, Oscar Kuang-Sheng Lee

**Affiliations:** 1Department of Medical Education, National Taiwan University Hospital, Taipei 100, Taiwan; twkao315@gmail.com; 2Faculty of Medicine, National Yang-Ming University, Taipei 100, Taiwan; 3Institute of Biophotonics, National Yang-Ming University, Taipei 112, Taiwan; aechiou@ym.edu.tw; 4Biophotonics and Molecular Imaging Research Center, National Yang-Ming University, Taipei 112, Taiwan; 5Institute of Physics, Academia Sinica, Taipei 115, Taiwan; khlin@phys.sinica.edu.tw; 6Department of Physiology and Pharmacology, Chang Gung University College of Medicine, Taoyuan 333, Taiwan; lyspub1@gmail.com; 7Department of Plastic and Reconstructive Surgery, Chang Gung Memorial Hospital, Taoyuan 333, Taiwan; 8Institute of Clinical Medicine and Stem Cell Research Center, National Yang-Ming University, Taipei 115, Taiwan; 9Department of Medical Research, Taipei Veterans General Hospital, Taipei 115, Taiwan; 10Department of Orthopedics, School of Medicine, China Medical University, Taichung 404, Taiwan; 11Department of Orthopedics, China Medical University Hospital, Taichung 404, Taiwan

**Keywords:** mesenchymal stem cell, osteogenesis, viscoelasticity

## Abstract

Human mesenchymal stem cells (hMSCs) possess potential of bone formation and were proposed as ideal material against osteoporosis. Although interrogation of directing effect on lineage specification by physical cues has been proposed, how mechanical stimulation impacts intracellular viscoelasticity during osteogenesis remained enigmatic. Cyto-friendly 3D matrix was prepared with polyacrylamide and conjugated fibronectin. The hMSCs were injected with fluorescent beads and chemically-induced toward osteogenesis. The mechanical properties were assessed using video particle tracking microrheology. Inverted epifluorescence microscope was exploited to capture the Brownian trajectory of hMSCs. Mean square displacement was calculated and transformed into intracellular viscoelasticity. Two different stiffness of microspheres (12 kPa, 1 kPa) were established. A total of 45 cells were assessed. hMSCs possessed equivalent mechanical traits initially in the first week, while cells cultured in rigid matrix displayed significant elevation over elastic (G′) and viscous moduli (G″) on day 7 (*p* < 0.01) and 14 (*p* < 0.01). However, after two weeks, soft niches no longer stiffened hMSCs, whereas the effect by rigid substrates was consistently during the entire differentiation course. Stiffness of matrix impacted the viscoelasticity of hMSCs. Detailed recognition of how microenvironment impacts mechanical properties and differentiation of hMSCs will facilitate the advancement of tissue engineering and regenerative medicine.

## 1. Introduction

Osteoporosis has been a longstanding issue in healthcare, and its prevalence is on constant rising due to aging of society and increased clinical awareness. According to recent epidemiological report, 10 million individuals over the age of 50 in the U.S. endorsed such diagnosis [1], and a substantial portion of this cohort was eventually complicated with a compression fracture. Mechanistically, prolonged phase of bone absorption by osteoclasts and shortened duration of bone formation by osteoblasts brings about pathological bone remodeling [2]. Wolff’s law suggested the strength of bone reflects mechanical loading. Specific population under abnormal physical pressure, e.g., bed-ridden individuals and astronauts, are therefore predisposed for osteoporosis. Traditional approach, in addition to a rehabilitation program, consisted of medical treatment including as biphosphates, selective estrogen receptor modulators, parathyroid hormone analog, monoclonal antibodies such as denosumab and romosozumab, as well as vitamin D and calcium supplementation [3]. Despite the demonstration of therapeutic efficacy by the mentioned drug noted above, clinically poor responders are still frequently encountered. Development of newer treatments for osteoporosis is needed to better manage osteoporosis.

Bone reconstruction by employing human mesenchymal stem cells (hMSCs) has currently become an alternative. To direct the lineage commitment of hMSCs, literature suggested not only biochemical induction but physical stimuli also ought to be fine-tuned. Studies demonstrated even in the absence of pharmaceutical cues, the rigidity of culture microenvironment impacts cell fate choice [4], as stiff and soft niche respectively stimulates hMSCs toward osteogenesis and adipogenesis. Specifically, high compliance of extracellular matrix influenced MAPK signaling by activating Rho/ROCK and localizing YAP/TAZ into the nucleus, thereby fastening RUNX2 to promote osteogenesis [5,6]. Various parameters of the matrix, including thickness and cross-linking, were proposed to influence lineage specification of indwelling stem cells [7]. Furthremore, dimensionality was proposed to be the crucial parameter for induction of pluripotency and differentiation [8,9]. Hsieh et al. also suggested in conjunction with matrix stiffness, the dimensionality synergistically orchestrated the expression of osteogenic marker genes of hMSCs [10]. Rather than conventional culture method, three dimensional scaffolds more precisely recapitulated the in vivo condition. This biomimetic niche facilitated the advancement of tissue engineering [11], and the optimized spherical boundary condition offered the platform to investigate cell spreading, migration, and morphogenesis.

Additionally, documentation of the mechanical properties during cell fate commitment is the cornerstone to monitor the progression of osteogenesis. Young’s modulus, defining correlation between stress and strain, gradually escalates as the bone formation matures. For most circumstances, the intracellular mechanical properties were assessed by the atomic force microscope (AFM) [12]. However, such measure necessitated the contact of probe tip and cell of interest, which inevitably altered the rigidity of cytoskeleton. Since the hMSCs viscoelasticity was in inverse proportion to the total Brownian motion distance of intracellular particles, video particle tracking microrheology (VPTM) methodology and in conjunction with the generalized Einstein-Stokes equation were exploited to indirectly evaluate the mechanical properties [13].

The aim of this study was to interrogate the mechanical impact of culture niche on hMSCs during osteogenic differentiation, and thereby established the protocol for future investigation on mechanotransduction. Depiction of the study design was exhibited in Figure A1.

## 2. Results

### 2.1. Scaffold Profiling and Bead Concentration

3D porous scaffolds with two distinct rigidities were manufactured for hMSCs culture. Stiffness at 1 kPa and 12 kPa were opted to physiologically recapitulate the mechanical properties of the nervous system and mineralized bone. Pore size of the matrices was controlled uniformly at a 100 μm diameter for optimal osteogenesis (Figure 1). The initial concentration of fluorescent beads was 6.2 × 10^12^ particle/mL. Although a 1/20 dilution ratio is widely applied in 2D culture setting, the 3D substrate with additional z axis inevitably result in the projection overlapping the fluorescent beads and thus lead to misrecognition of the random walk for individual particles. To fine-tune the optimal ratio for dilution, three arbitrary values, 1/20, 1/30, and 1/50, were examined. Those cells subjected to the injection of 1/20 and 1/30 diluted particles were too crowded to analyze individual behavior of the beads, whilst the 1/50 mixing ratio was eligible to both prevent excessive particle-to-particle disturbance and offer sufficient tracking information (Figure 2). Therefore, all trials in the study were based on this dilutive condition.

### 2.2. Postures of hMSCs in Porous Scaffolds

Unlike 2D substrates where cultured cells flattened universally, the indwelling hMSCs in 3D scaffolds assumed heterogenous postures, e.g., bridging over (Figure 3A) or vertical lying (Figure 3B). The most frequently encountered morphology was pyramidal-shaped (Figure 3C), of which respectively coplanar but not linear anchoring forces at 3 edges reached static equilibrium as Lami’s theorem suggested. An equal number of cells with each morphology were collected for calculation. To study whether or not bone formation would be influenced by fluorescent beads, the expressions of osteogenic markers, Osterix and Runx2, in those cells status post particle injection were quantified and compared with control. Initial elevation of the markers corresponded well with the course of osteogenesis. Additionally, the introduction of fluorescent beads did not bring about a significant difference throughout the differentiation course (Figure 4).

### 2.3. Assessment of Intracellular Viscoelasticity by VPTM

A total of 45 different cells in three independent trials were assessed using confocal microscope. 4 to 10 fluorescent particles per hMSCs were identified for analysis, and every hMSC of interest was snapshotted with both bright field and confocal resolution. Each Brownian motion was traced, and the trajectory was projected to transverse xy-plain for subsequent calculation. The motion was depicted in respective to time and along x/y axis individually. Integration thereof yielded mean square displacement, which was applied in the function of time step to generalized Einstein-Stokes equation. (Figure 5). For stable and adequate outcome, the frequency was set as 10 Hz in the equation to obtain complex shear moduli.

Initially at Day 0, the mechanical properties of hMSCs in both 1 kPa and 12 kPa group was comparable, and no significant differences in elastic modulus were found by the first week after chemical induction toward osteogenesis. Viscosity of hMSCs in 1 kPa scaffold, however, was up to 168 Pa and was slightly higher than that in 12 kPa substrate, 135 Pa. Interestingly, hMSCs in the 12 kPa scaffolds exhibited continuously strong increase in both elastic as well as viscous moduli, i.e., G′ and G″, 2 weeks (192 Pa) and 3 weeks (204 Pa) post-induction. As for the effect posed by soft 1 kPa matrix, the G′ and G″ fluctuated to (114 Pa, 169 Pa) and (129 Pa, 161 Pa), respectively (Figure 6A,B). Therefore, rigid matrix not only biased hMSCs toward bone formation but also elevated the rigidity of the cytoskeleton.

Furthermore, to appreciate the trend of alterations, follow-up measurements were put into comparison with Day 0 initial value in respective groups. Intriguingly, soft 1 kPa niches barely exerted any further effect on viscoelasticity enhancement after Day 7. Rigid 12 kPa matrices, on the contrary, promoted the escalation of both G′ and G″ up to 204 Pa and 222 Pa in a consistent manner during the entire osteo-specification course (Figure 6C,D), illustrating hMSCs perceived the varied degree of the mechanical impact during bone formation by exhibiting different trends of changes on viscoelasticity in response.

## 3. Discussion

Although it has been well-established that the mechanical properties of residing microenvironment wound bias lineage specification of hMSCs [14], how the architecture of cytoskeleton responded to physical stimuli remained enigmatic. In this study, the culture platform of 3D polyacrylamide scaffold was established, and the alteration of intracellular viscoelasticity was assessed in conjunction with VPTM employment. Osteoblast maturation has been proposed to accompany with the elevation of viscoelasticity, and the substrate stiffness modulated the behavior of osteoblasts [15]. Our study illustrated the viscous and elastic modulus of hMSCs in 12 kPa substrates was on constant increasing in the entire osteogenic course. On the contrary, Young’s modulus of those in 1 kPa reached plateau by the second week post chemical induction. 12 kPa was hence the preferable rigidity of 3D substrates for residence to 1 kPa in terms of both osteolineage commitment and cytoskeleton stiffening. Culturing cells in stiffer 3D scaffolds prior to engraftment ensured the mechanical properties would rise to level of osteoblast and potentially rescue osteoporosis.

To accurately quantify mechanical characteristics, minimization of unintended physical impacts during cell transfer and property measurement was of priority. Aside from cryopreservation so as to halt confounding mechanotranductive effects, traditional AFM was replaced for the assessment of hMSCs mechanical characteristics. To ideally evaluate spatiotemporal rheological properties of hMSCs, the passive microrheology technique, VPTM [16,17], was firstly employed to study Young’s modulus during osteogenesis. Soft tissue was especially feasible and already ubiquitous for VPTM measurement. Using diffusing wave spectroscopy, Chenet et al. quantified the viscoelasticity of synovial fluid of arthritis patient [18]. Chen et al. applied VPTM to interrogate viscoelasticity of HeLa cells during cell division [19]. However, each sample required the establishment of individualized protocol and study design. In this study, we pinpointed the dilution ratio of 1/50 for fluorescent particles with phosphate buffered saline, and also assigned the frequency parameter at 10 Hz for transformation. The protocol could be applied to other hMSCs experiments, while customized modifications might be necessary regarding the study objectives.

As the advancement of tissue engineering and biophysical materials, the concept on 3D matrix for cell indwelling was introduced [20]. Adapted from the experience of fibroblast culture, the negatively curved surface promoted cell adhesion and realized an extra-dimensionally to interrogate cellular behavior. Moreover, the spherically boundary condition was demonstrated to impact hMSCs osteogenesis. Enlightened by previous study, 3D substrates with a diameter at 100 or 150 μm were the preferred size for bone formation [21]. In this study, the material for matrix manufacturing was designed universally at 100 μm to boost osteogenesis. Previous study pointed out the rigidity of 3D substrate at 11~30 kPa influenced hMSCs commitment by orchestrating integrin binding and adhesion ligands [22]. Literature also highlighted in 3D setting, the cellular behavior and ligand chemistry responding to substrate stiffness were different [23]. The dissimilarity between topological structures lies in the mechanical cues the niche imposed on the cells within [24]. 2D collagen-coated microenvironment brought about apical-basal polarity, whereas 3D niche was compatible with unconstrained spreading. Interestingly, recent evidence linked the using of 3D matrix as culture condition to the assistance of stem cell reprogramming. The scaffold prepared by 3D printing was also proposed to improve bone regeneration by bone marrow-derived mesenchymal stem cells [25], and 3D extracellular matrix was illustrated to both offer biophysical stimuli but also preserve tissue [26]. These together underscored the effect of an “extra-dimension” on stem cell physiology.

There are limitations of this study. First, the mechanical background hMSCs experienced before indwelling into the 3D spheres was undetermined. 3D-to-3D transfer was techanically challenging and the protocol was still immature. Egger et al. performed the first isolation of adipocytes from native tissue directly into 3D hydrogel without contacting 2D dish [27]. In our experiments, hMSCs were maintained on hard 2D plastic surface before transferred to 3D culture, thereby resulting in mechanical memory. This confounding parameter might interfere cell culture. Manipulation of miR-21 was proposed to erase this effect [28] but no references were available for execution under the circumstance of hMSCs osteogenesis. Second, the mechanical stimulus cultured hMSCs underwent depended on its posture. Topologically different anchorage morphologies detected respective physical cues and thus transduced non-identical signals for proliferation and fate commitment [29]. Even though VPTM was indicated to the single cell level, opted cells of interest were selected manually according to visual selection over cellular morphology and therefore subjective. Development of autonomic calculation and grouping by physical characteristics was mandated for stratified analysis of the viscoelasticity.

As for future prospective, the mechanotransductive pathway for hMSCs is not fully understood. Different types of mechanical loading have been proposed, including shear stress, compression, vibration, etc. Nevertheless, how the cells adapted under various physical signals and the changes in marker gene expression were rarely touched by previous studies. Becquart et al. reported intermittent shear stress and cyclic hydrostatic pressure influenced the production of nitric oxide expression of mechanosensitive genes, thereby harnessing stem cell differentiation [30]. Moreover, the mechanical feedback of cultured hMSCs to the local niche for cell fate manipulation remained enigmatic. Detailed appreciation on the mechanical impacts for hMSCs differentiation is the cornerstone for the development of tissue engineering and materials for regeneration.

## 4. Materials and Methods

### 4.1. Fabrication of Polyacrylamide 3D Scaffolds

Polyacrylamide matrices, unlike gelatin scaffold with otherwise unchangeable rigidity, were manufactured. Fundamental ingredients were acrylamide (Bio-red, Hercules, CA, USA) as monomer and bis-acrylamide (Bio-red, Hercules, CA, USA) as crosslinker. The mixing ratio was [18.75%, 1.13%]/[7.00%, 0.20%] for stiff/soft matrices, and former AFM measurements determined the rigidity at 12 kPa and 1 kPa, respectively. Other constituents, including gelling initiator ammonium persulfate (Sigma-Aldrich, St. Louis, MO, USA), surfactant Pluronic F-127 (Sigma-Aldrich, St. Louis, MO, USA), and catalyst N,N,N′,N′-tetramethylethylenediamine (Sigma-Aldrich, St. Louis, MO, USA), accounted for 1% of total mixture. Besides, soft-lithographed polydimethylsiloxane-based microfluidic device was exploited to direct the flow and facilitate the mixing of input substances. Polyacrylamide, ammonium persulfate, and nitrogen gas saturated with perfluorohexane (C6F14) were introduced by three individual inlets, and monodisperse foams to fulfill reservoir (6 mm in width, 1 mm in height) were collected at the conjoint outlet. The semi-solid yield was solidified in 80 °C oven and degassed in vacuum to patent homogeneous 100 μm pores. Scaffolds were eventually treated with 1 mg/mL sulfosuccinimidyl 6-(4′-azido-2′- nitrophenylamino) hexanoic acid (Thermo, Waltham, MA, USA) and activated under 8-min ultraviolet radiation to realize the conjugation of 0.5 mg/mL fibronectin (Sigma-Aldrich, St. Louis, MO, USA) for cyto-friendly culture condition. Summary of utilized techniques was listed (Table 1).

### 4.2. Injection and Tracking of Fluorescent Particles

To prevent confounding mechanical influences and for execution convenience, the fluorescent beads were injected into hMSCs prior to seeding. The protocol was adapted from preceding reports. Through the propulsion of helium, the biolistic delivery system was applied to introduce 0.2 μm red fluorescent carboxylate-modified microspheres. The particle was negatively-charged and possessed maximum emission at the wavelength of 605 ± 5 nm. These beads randomly walked around the cytoskeleton under otherwise minimal exogenous stimuli except the rigidity of indwelled matrix. The Brownian displacement of fluorescent particles were captured using inverted epifluorescence microscope equipped with charge-coupled device camera and oil immersion objective lens (numerical aperture =1.45, 100×).

### 4.3. hMSCs Culture, Maintenance and Chemical Induction

Commercially available hMSCs with passage number 14 were utilized in this study (Steminent Biotherapeutics, Taipei, Taiwan). Cells were cultured in growth medium with the formation of Iscove’s modified Dulbecco’s Medium (IMDM, Invitrogen, Carlsbad, CA, USA), 10% fetal bovine serum (Sigma-Aldrich), 10 ng/mL basic fibroblast growth factor (Sigma-Aldrich, St. Louis, MO, USA), 100 units of penicillin, 1000 units of streptomycin, and 2 mM L-glutamine (Sigma-Aldrich, St. Louis, MO, USA). The incubation condition was constantly set as 37 °C, 5% carbon dioxide. Initially hMSCs were seeded at 30% confluent. After a week of expansion to reach the confluency of 80%, the cells were detached by Trypsin-EDTA (Sigma-Aldrich, St. Louis, MO, USA). The cells were than harvested and introduced to 3D scaffolds with the density of 1.5 × 105 of cells per matrix after two passages of culture. Such timing was defined as day 0.

For biochemical induction toward bone formation, the original medium was substituted with osteogenic regimen 1 day after hMSCs were administered on 3D matrices. The recipe was IMDM-based, supplemented with 0.1 μM dexamethasone (Sigma-Aldrich, St. Louis, MO, USA), 10 mM ß-Glycerol Phosphate (Sigma-Aldrich, St. Louis, MO, USA), 1% penicillin-streptomycin- glutamine (Gibco BRL, Gaithersburg, MD, USA), and 0.2 mM autologous serum-derived albumin (Sigma-Aldrich, St. Louis, MO, USA). The medium was provided for another 21 days and changed twice a week. The hMSCs were harvested on day 0 and 7, 14, 21 days post-induction for further assessments.

### 4.4. RNA Isolation and Quantitative Real-Time Polymerase Chain Reaction

Total RNA was extracted from cultured hMSCs with and without fluorescent bead injection on day 0 as well as 7, 14, 21 days post-induction by RNeasy Mini Kit (QIAGEN, Manchester, UK). Initially, lysis buffer, RLT solution with 1% ß-mercaptoethanol, and commercial kits were employed to isolate RNA means as usual. Yet due to excessive polyacrylamide debris of 3D scaffolds on the filtering membrane, the efficacy of RNA harvest was compromised and the purity was suboptimal. In response, trizol method was adapted. Chloroform was added after smashing the matrix with pipetman tips. polyacrylamide residues were aggregated at the interface of supernatant/organic layer after accelerated centrifugation. Subsequent double washing with alcohol further assured the immaculacy of RNA pallets. To reach adequate amount, the RNA was juiced from 1 slice of 2D hydrogel/24 pieces of 3D PA scaffolds per condition. 1 μg of isolated RNA was reverse transcribed to complementary DNA using MMLV High Performance Reverse Transcriptase (EPICENTRE, Madison, WI, USA). Real-time polymerase chain reaction for quantification was accomplished with TaqMan Fast Universal PCR Master Mix. Two osteogeneic markers, Osterix and Runx2, were assessed.

### 4.5. Statistical Analysis

All exhibited statistics were presented as mean ± standard error of mean and obtained in triplicate for statistical validation. three independent experiments. Quantitative results were analyzed by SPSS software (Version 19.0, IBM, Armonk, New York, USA) and compared by Student *t*-test. *p* value < 0.05 was considered significant and with (*) denotation, while (**) represented *p* < 0.01.

## 5. Conclusions

The study pointed out the alteration of 3D matrix rigidity would orchestrate the mechanical parameters of cultured hMSCs. By our established platform for substrate manufacturing and VPTM protocol to assess Young’s modulus, the 12 kPa but not 1 kPa matrix significantly facilitated the stiffening of hMSCs throughout the 21-day chemically-stimulated osteogenic course. Such characteristics may be further adapted for identification of osteogenic maturity in hMSCs, and may serve as a readout for the development of osteoinductive biomaterials.

## Figures and Tables

**Figure 1 ijms-22-02441-f001:**
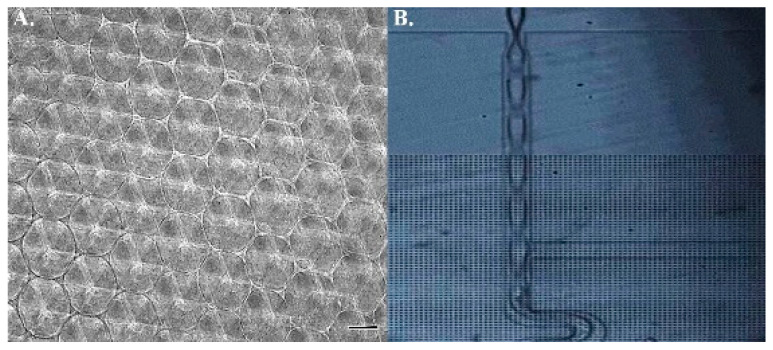
Fabrication of 3D polyacrylamide scaffold. (**A**) The diameter of each sphere was set at 100 µm. (**B**) The microfluidic device was employed to carry nitrogen for polyacrylamide and ammonium persulfate mixing. Scale bar = 100 µm.

**Figure 2 ijms-22-02441-f002:**
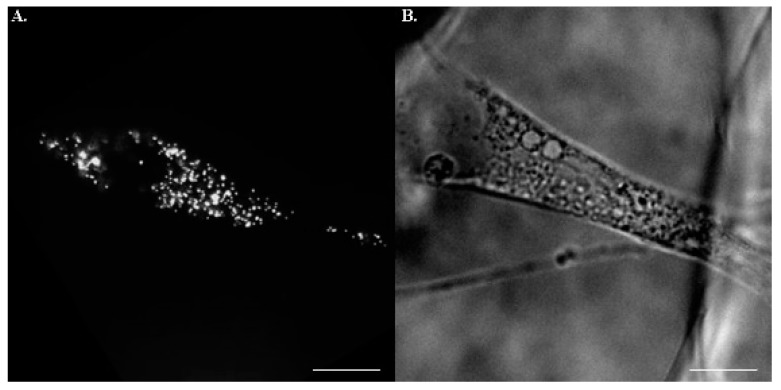
An example of fluorescent beads in hMSCs. Under (**A**) TxRed and (**B**) bright field illustration with a charge-coupled camera, the intracellular fluorescent particles were readily identifiable. Scale bar = 20 µm.

**Figure 3 ijms-22-02441-f003:**
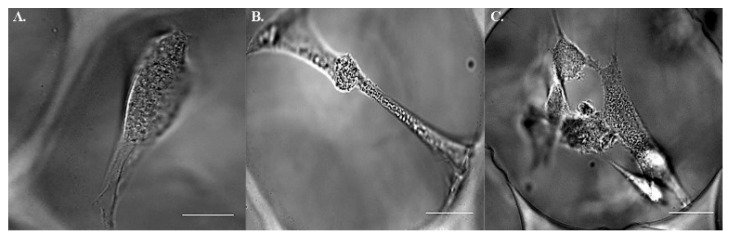
Heterogeneous morphologies of hMSCs indwelling 3D substrates. Bright field of hMSCs (**A**) vertically lying (**B**) bridging over (**C**) pyramidal-shaped. Scale bar = 20 µm.

**Figure 4 ijms-22-02441-f004:**
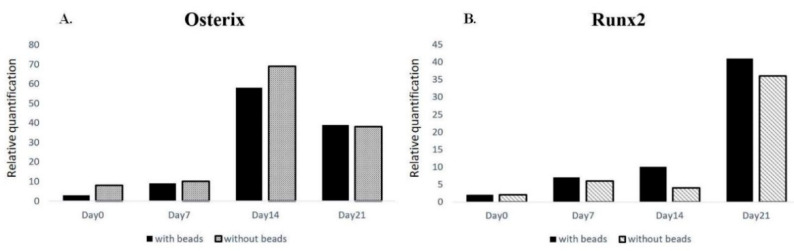
Injected fluorescent beads did not influence hMSCs osteogensis. No significant difference was present in marker gene expressions, (**A**) Osterix and (**B**) Runx2, in cells with or without fluorescent particles.

**Figure 5 ijms-22-02441-f005:**
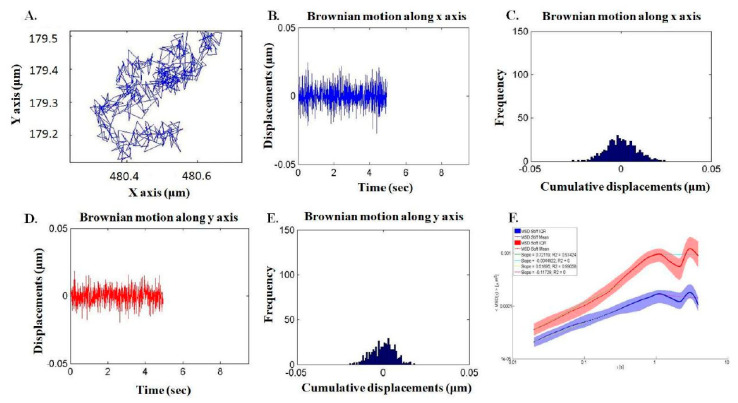
Computation of the hMSCs mechanical properties by passive microrheology. (**A**) The Brownian motion of beads in hMSC of interest was traced and projected to XY-plain. (**B**,**D**) In respect to time, the motion along x and y axis was depicted individually. (**C**,**E**) All documented displacements were integrated and expressed with frequency to calculate mean square displacement (MSD). (**F**) Young’s modulus was yielded by applying MSD into generalized Einstein-Stokes equation.

**Figure 6 ijms-22-02441-f006:**
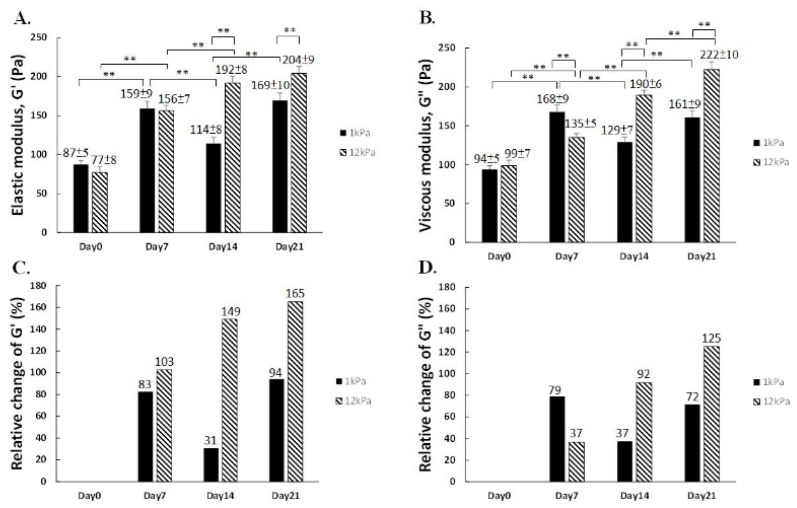
Viscoelasticity of hMSCs during osteogenesis. (**A**,**B**) The elastic (G′) and viscous modulus (G″) of hMSCs cultured in stiff and soft substrates. (**C**,**D**) Relative changes of Young’s modulus in the course of chemically-induced osteogenesis. (** *p* < 0.01)

**Table 1 ijms-22-02441-t001:** List of the techniques utilized in the study. hMSC: human mesenchymal stem cell.

Technique	Purpose	Application in this Study
Polydimethylsiloxane-based microfluidic device	To propel the movement of those ingredients introduced for scaffold manufacturing	The device was employed to facilitate mixing of input materials
Biolistic delivery system	To place particle into cell of interest	Fluorescent beads were injected intracellularly to hMSCs
Charge-coupled camera	To capture cellular image under designated resolution	TxRed and bright images of hMSCs were obtained.
Video particle tracking microrheology	To deduce Young’s modulus from Brownian motion of intracellular particles	The viscoelasticity of hMSCs were yielded by tracing the random walk of fluorescent particles

## Data Availability

The dataset supporting the conclusions of this article is included within the article.

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
