# Peer review of "Alteration of 3D Matrix Stiffness Regulates Viscoelasticity of Human Mesenchymal Stem Cells"

_ijms, 2021, doi:10.3390/ijms22052441_

Round 1
Reviewer 1 Report
In this manuscript Kao TG et al present how matrix stiffness regulates the mechanical properties of hMSCs. The authors used different polyacrilamide gels and injected fluorescent beads into hMSCs to determine their mechanical properties by passive microrehology.
In general, the figures are too simple and not well self-explained (i.e. Figure 4, how did the author analyse Runx2/Osterix expression?). It looks too preliminary in my opinion. Furthermore when it has been broadly studied how matrix stiffness might impact on the mechanical properties of hMSCs. Although I agree with microrehology is an interesting technique compared to AFM, I can't find any remarkable advances in the field.
Other comments:
There is no quantification on the cell morphologies (Figure 3). Authors should show more than one cell and the quantification. Could they perform confocal analysis to present 3D reconstructions of the morphologies? Showing other kind of IFs instead of brightfield might be more resolutive.
In my opinion Figure 5 is the most important of the article. However, the authors should show a movie or cell reconstruction showing a representative cell of the passive microrehology experiment.
Reviewer 2 Report
Kao et al., present interesting work. The manuscript requires extensive revision to increase clarity.
Main issues:
- The experimental design should be represented in a diagram.
- Each technique used and what to expect with interpretation should be summarized in a Table.
- Please add legend for each figure/Table.
Reviewer 3 Report
The current manuscript aims at investigating how alteration of 3D matrix stiffness regulates viscoelasticity of hMSCs. The idea of the study is very original and has a great potential of providing a significant advance of current knowledge in regenerative medicine. Introduction is adequate and provides a solid motivation for the research, while also showing the background and recent progress in the field. The aim of the study is well stated. Materials and methods are very complex, well-chosen, while the study design is appropriate and well-described, making this work easily reproducible. Results are well-presented, with adequate figures and statistical analysis. Discussions are very well conducted and point out the originality and added value of this study, while comparing it with other connected works. Conclusions are sound and in accordance with the results. English language is adequate throughout the whole manuscript. My opinion is that this represents a valuable work, presenting a complex study, with great potential for regenerative medicine. Therefore, I recommend publication as it is.
Author Response
Thank you for the comments. We will adjust the content based on other reviewers' opinions.
Reviewer 4 Report
The authors analyzed the effect of 2 3d models with a different stiffness on the physical properties of MSCs under differentiation, with a focus of a possible improvement of osteoporosis treatment.
The paper is interesting, but it seems very preliminar, in terms of a possbile improvement of MSC differentation.
The authors seed 1,5x105 cells/scaffold but then analyzed 45 cells/scaffold with confocal microscope, why so few cells?
The authors analyzed the effect of stiffeness on MSCs during their differentation, however the expression of just 2 markers is not sufficient to claim a differentation, other techniques should be included to really verify the differentation. Moreover, the marker expression in fig 1 did not correlate with the change they reported (i.e. Osterix at day14 with parameters in fig.6.
Round 2
Reviewer 1 Report
Authors replied my issues and I have no further suggestions to do.
Author Response
We appreciate your comment.
Reviewer 2 Report
The authors have revised the manuscript as requested. The new Tables should be in the text.
Author Response
We appreciate your comment. The table has been adjusted from supplementary to main text table 1.
Reviewer 4 Report
I can only confirm my observations reported yesterday. The authors reply to my comments, I can accept the reply to the first issue (about the number of cells), but I still consider too preliminar a study that report the change of physical properties of MSCs (albeit interesting!) without any strong correlation with their differentation. Moreover, they did not answer to the last issue, about the lack of correlation between marker expression in fig 1 andh the change they reported (i.e. Osterix at day14) in fig.6.
Author Response
Thank you for the insightful comments.
First, we highly appreciate your recognition regarding our elucidation upon stem cell viscoelasticity during osteogenesis.
Second, in figure 1 we illustrated our apparatus for scaffold manufacturing. Besides, the purpose of us to present gene expression data in Fig 4 is merely to affirm that the injection of fluorescent beads intracellularly, as compared with those in which no particle was administered, does not influence the course of differentiation. Since at this stage it is still technically-challenging to harvest optimized quality and quantity of hMSCs DNA from our 3D polyacrylamide scaffold, the qPCR outcome, albeit able to reflect the general course of lineage specification, might not serve as exact numerical documentation of marker gene expressions. Therefore, highlight of this study was that we attempted to compare the alteration of mechanical properties secondary to different rigidity of cultured matrix in Fig 6. We look forward to your understanding that the improvement of culture platform and genetic investigation is currently in progress. Detailed investigation of corresponding genetic mechanism will be reported in future study.
Thank you again for the kind suggestions.